# MOFs and MOF-Derived Materials for Antibacterial Application

**DOI:** 10.3390/jfb13040215

**Published:** 2022-11-03

**Authors:** Xin Zhang, Feng Peng, Donghui Wang

**Affiliations:** 1School of Materials Science and Engineering, Hebei University of Technology, Tianjin 300130, China; 2Medical Research Center, Department of Orthopedics, Guangdong Provincial People’s Hospital, Guangdong Academy of Medical Sciences, Guangzhou 510080, China; 3School of Health Sciences and Biomedical Engineering, Hebei University of Technology, Tianjin 300130, China

**Keywords:** metal organic framework (MOF), MOF-based composites, biocompatibility, antibacterial

## Abstract

Bacterial infections pose a serious threat to people’s health. Efforts are being made to develop antibacterial agents that can inhibit bacterial growth, prevent biofilm formation, and kill bacteria. In recent years, materials based on metal organic frameworks (MOFs) have attracted significant attention for various antibacterial applications due to their high specific surface area, high enzyme-like activity, and continuous release of metal ions. This paper reviews the recent progress of MOFs as antibacterial agents, focusing on preparation methods, fundamental antibacterial mechanisms, and strategies to enhance their antibacterial effects. Finally, several prospects related to MOFs for antibacterial application are proposed, aiming to provide possible research directions in this field.

## 1. Introduction

A significant factor inducing human sickness and death has always been disease-causing microbes (bacteria, fungi, and viruses) [1,2,3,4,5,6]. Alexander Fleming, in 1928, discovered that penicillin was successful at preventing bacterial colonization, and researchers since then have discovered a range of drugs for antibacterial treatment [7,8]. However, drug-resistant bacterial populations have emerged as a result of long-term use of antibiotics. This poses a significant threat to human health [9,10,11], so there is an urgent need to find alternatives to antibiotics.

Many antibacterial materials, including metal-based (e.g., copper, silver, zinc) nanomaterials [12,13,14], organic materials [15,16,17,18], semiconductor photocatalytic materials [19,20,21], and natural biological antibacterial materials [22], have been developed and used in the field of biomedicine in recent years. However, most of them have not achieved clinical application. Metal-based nanomaterials and organic materials can kill bacteria by releasing toxic metal ions or antibacterial molecules, but the abrupt release of active substances has a short duration of efficacy and can be harmful to organisms [12,13,14,16,18]. Semiconductor photocatalytic materials can absorb externally stimulated light and generate photogenerated carriers to stimulate enhanced enzyme-like activity to kill bacteria. However, their limited light absorption capacity leads to low catalytic activity and an unremarkable antibacterial effect [19,20]. Natural biological antibacterial materials are harmless substances extracted from plants and animals, which can be slightly modified or even directly applied for antibacterial therapy in organisms, but the few available sources, expense, and complex process used to obtain these materials limit their wide clinical application. Furthermore, there are some prospective biomimetic technological routes to construct antibacterial drugs (e.g., the protein cages, and cell-membrane-camouflage). Ferritin has reversible self-assembly properties and has been used to construct artificial protein nanocages. Ferritin-based protein has nanocage-like structures with endowed chambers that provide sites for the encapsulation of antibacterial drugs. Furthermore, they can effectively target bacteria because of their unique positively charged nature. However, protein nanocages are known to be nanoscale, which can only encapsulate smaller molecule drugs and are limited in large molecule loading, and the existing protein encapsulation technology is not sufficient to precisely encapsulate functional substances [23]. Drugs camouflaged by various cell membranes can take advantage of the complex biological components and functions of cell membranes to successfully “trick” the body’s immune system, smoothly passes immune recognition, prolongs circulation time in the blood, and enhances the efficiency of drug delivery at the lesion site. Although the coupling of cell membrane-coated technology with antibacterial drugs has showed satisfactory results in terms of screening ability from natural products, the screening efficiency still could not meet the requirements of practical applications. Additionally, its stability still needs further improvement to adapt to the complex physiological environment [24,25]. MOFs and MOF-derived materials stand out in the antimicrobial field because of their designable structure and adjustable size.

In 1995, Yaghi et al. [26] synthesized a two-dimensional coordination compound structure called a metal organic framework (MOF, please refer to Abbreviations at the end of this review) using a rigid organic ligand, BTC, and a transition metal ion, Co^2+^. MOFs are hybrid inorganic–organic materials with pores. They are formed by bonded self-assembly of metal centers (metal ions or clusters) and organic ligands (bipedal or multi-pedal), and MOFs with desirable structures and functions can be obtained by rational design of IBUs and OBUs. As soon as this concept was proposed, it was developed at an astonishing pace in the following two decades. 

Nowadays, numerous types of MOFs have been developed; the most common include IRMOF, ZIF, MIL, CPL, UIO, and PCN. IRMOF is an octahedral microporous crystalline material self-assembled from [Zn_4_O]^6+^ with a series of aromatic carboxylic acid ligands [27]. ZIF is synthesized by the reaction of Zn^2+^ or Co^2+^ with imidazole and exhibits a zeolite-like structure [28,29]. MIL is synthesized by transition metal ions with dicarboxylic acid ligands (succinic acid, glutaric acid, etc.) [30]. CPL consists of six metal ions coordinated to a neutral nitrogen-containing heterocycle [31]. The UIO MOFs are three-dimensional microporous materials formed by the coordination of [Zr_6_O_4_(OH)_4_] with BDC, which contains an octahedral central pore cage and eight tetrahedral corner cages [32]. PCN contains multiple cuboctahedral nanopore cages, which can form a cage–pore channel-like topology in space [33]. The above MOFs have been extensively studied and applied in the fields of adsorption and separation, energy storage, sensing, catalysis, pollutant removal, electrochemistry, and biomedicine [29,34,35,36,37,38,39,40,41,42]. In addition, there are many new types of MOFs (HKUST, etc.), which have also been widely studied in recent years and are gradually being applied in various fields [43,44].

In the field of biomedicine, MOFs have emerged as an ideal material for various antibacterial applications because of their preferable functions, such as controlled/stimulated decomposition, strong interaction with bacterial membranes, ROS production under irradiation, and high loading amount and controlled release of other antibacterial agents [45]. A large number of studies and reviews appeared in this field focusing on the composition, structure, and antibacterial mechanisms of different MOFs, and showed that MOFs and MOF-derived materials present a high bactericidal activity whose antibacterial rate exceeds 99% at appropriate doses (Table 1). [46,47,48]. However, the possible cytotoxicity limited the clinic application of MOFs and MOF-based materials, and it is still a great challenge to enhance their bactericidal activity and biocompatibility simultaneously. In this review, we summarize the preparation and antibacterial mechanisms of MOFs. Based on recent representative works on the development of MOFs, the possible mechanisms in antibacterial applications can be categorized as physical interaction, component release, CDT, PDT, PTT, SDT, and synergistic therapy. Particularly, a comprehensive review of the strategies to enhance the antibacterial effect of MOFs is presented in terms of both dynamic and thermodynamic aspects, including modulation of the size, pore size, and coordination environment of active sites, and the construction of MOF-based composites. Finally, the prospects and challenges of MOF-based antibacterial materials are discussed.

## 2. Preparation Methods of MOFs

There are many methods to prepare MOFs. The same type of MOFs or even the same MOFs can be obtained by different preparation methods, which will confer different properties. In this section, we introduce several commonly used methods of synthesizing MOFs: one-pot, hydrothermal (solvent thermal), ultrasonic, electrochemical, and mechanochemical methods (Figure 1); we summarize their advantages and disadvantages for researchers to choose the appropriate preparation method based on their requirements.

### 2.1. One-Pot Method

One-pot synthesis is a process in which the precursors are co-mixed in a solvent and reacted under stirring. Wang et al. [65] synthesized Zr-based MOFs (Zr/UIO-66-COOH) using Zr^4+^ as the metal centers and H_2_BDC as the organic ligand in the CF_3_COOH solution. Huang et al. [66] synthesized Zn-based MOFs (MOCP) with a yield of 90% using Zn(NO_3_)_2_ and H_2_BDC as reactants in DMF solution with TEA at room temperature.

The advantage of this method is that the cost is low, the production is large, and the experimental conditions can be easily achieved. In addition, it is a highly controllable process, since researchers can add reactants at any time during the reaction. However, synthesized MOFs usually contain impurities. Therefore, the one-pot method is not suitable for applications that require high purity.

### 2.2. Hydrothermal (Solvent Thermal) Method

Hydrothermal (solvent thermal) synthesis is a process in which primitive mixtures are reacted in a closed system, such as an autoclave, using water or organic solvents at a certain temperature and with autogenous pressure. Li et al. [39] synthesized MOF-5 using Zn(NO_3_)_2_ and H_2_BDC as reactants in DMF by the solvent thermal method. The experimental data showed that the crystallinity of the product prepared by this method was higher compared to the one-pot method. This is because the high pressure under hydrothermal conditions increases the solubility of the precursors, consequently promoting the reaction and crystal growth. In addition, the synthesized MOF-5 had a specific surface area of up to 2900 m^2^ g^−1^ and could remain thermo-stable at 500 °C in the absence of moisture.

The hydrothermal (solvent thermal) method exhibits high specific surface area, high crystallinity, and thermal stability. However, this method is costly and cannot be well controlled, as reactants can only be added all at once. In addition, this method is not stable, since the specific pressure of the reactor, which is controlled by artificial tightening, is not a constant value, and this uncertainty will certainly affect the properties of the products.

### 2.3. Ultrasonic Method

Sonication is a process in which reactants are dissolved in solvent and continuously sonicated. Qiu et al. [67] first obtained Zn_3_(BTC)_2_ by sonication using an aqueous solution of Zn(CH_3_COO)_2_ and H_3_BTC dissolved in ethanol for 5 min. As the sonication time increased from 5 to 90 min, the size of the product increased from 50–100 to 700–900 nm, and the yield increased from 75.3 to 85.3%. Previous studies showed that mixing Zn(CH_3_COO)_2_ with H_3_BTC without ultrasound did not produce any product using the same reaction medium, temperature, and pressure. In theory, acoustic cavitation induced by the rupture of cavitation bubbles creates local hot spots with very high transient temperature (5000 K), pressure (1800 atm), and cooling rate (1010 K s^−1^) [68,69]. The region between the microbubble and the native solution has very large temperature and pressure gradients and rapid molecular motion, leading to the generation of an excited state, breaking of bonds, free radical formation, mechanical shock, and high shear gradients, so that the ultrasonic method allows some reactions to take place that are difficult to carry out by conventional methods.

Sonication is a low-cost and environmentally friendly method. The size and yield of MOFs synthesized by the ultrasonic method can be easily controlled. Moreover, some special reactions that cannot be carried out by conventional methods can be achieved because of the acoustic cavitation effect. However, the structure and purity of MOFs formed by sonication can vary.

### 2.4. Electrochemical Method

Electrochemical synthesis is a process in which monomers or compounds of different species and aggregates are prepared by electro-oxidation or electro-reduction in a conductive solvent. It is a widely used method to construct MOF films on substrates. Ameloot et al. [70] first synthesized Cu-based MOF (Cu_3_-(BTC)_2_) film on copper substrate by electrochemical deposition. The anode metal plates generated metal ions during electrolysis and then self-assembled with organic ligands on the electrode surface to form MOF films. They found that increasing the voltage from 2.5 to 25 V provided a higher concentration of metal ions near the electrode surface and produced a coating with smaller crystals, while adding water to the mixture slowed down the formation of crystals and formed a coating with larger crystals.

The advantage of the electrochemical method is that the synthesis process is simple and rapid, and the thickness of MOF films can be conveniently adjusted by adjusting the voltage and current applied and the electrolyte concentration. However, this method can only construct films on conductive substrates, limiting its wide application.

### 2.5. Mechanochemical Method

Mechanochemical synthesis is a process in which metal salts are reacted directly with organic ligands by mechanical grinding at a specific temperature. Katsenis et al. [71] synthesized a Zn-based MOF (ZIF-8) by the mechanochemical method using ZnO and 2-methylimidazole with small amounts of acetic acid or water as catalysts. They found that the mechanochemical process led to the collapse of the ZIF-8 structure. The higher the volume of solvent added during the grinding process, the more pronounced the amorphization of the product.

The mechanochemical method is environmentally friendly because of the low solvent volatilization. In addition, the amorphization of MOFs can be controlled. However, the crystallinity of the product synthesized by this method is low and the structure is easily destroyed in the process.

## 3. Antibacterial Mechanisms of MOFs

MOFs have a periodic coordination network in which metal centers and organic ligands are linked to each other and can be designed to have various typologies by regulating these two components [72,73]. In the field of biomedicine, several MOFs have been extensively studied for their preferable antibacterial activity resulting from their specific physical and chemical properties (such as slow release of metal ions or organic substances and enzyme-like, photocatalytic, photothermal, and ultrasonic activity) [49,74,75,76,77]. This section summarizes in detail the possible antibacterial mechanism of MOFs, which can be subdivided into physical interaction, metal ion release, organic ligand release, antibiotic load, gas load, CDT, PDT, PTT, SDT, and synergistic therapy (Figure 2 and Table 2).

### 3.1. Physical Interaction

Physical interaction refers to the process of direct interaction between materials with specific surface morphologies and bacteria in the physiological environment. Recently, some MOFs were synthesized with two-dimensional morphology that exhibited excellent antibacterial properties. Yuan et al. [53] synthesized ZIF-L with a nano-dagger surface, which is completely different from the dodecahedral structure of ZIF-8. The experimental data showed that ZIF-L exhibited excellent antibacterial activity (log reduction > 7 for *E. coli* and *S. aureus*) and sustainable bactericidal activity (more than four reapplications). They believed that the nano-dagger surface of ZIF-L could kill bacteria efficiently via physical interaction (Figure 3A). Wang et al. [54] synthesized four Cu-based MOFs (CPPs) with the same structure but different morphologies (rhombus layer, rhombus disk, rhombus lump, and bread-like morphology) by adding additives such as TEA and CH_3_COOH. The rate of deprotonation of organic ligands can be controlled by the addition of additives to further control particle size and shape. Among them, the bread-like and rhombus disk CPPs exhibited weak antibacterial activity (MIC > 50 μg mL^−1^), while the rhombus lump and rhombus layer CPPs showed good antibacterial activity against *B. subtilis*, *P. vulgaris*, *S. aureus*, *P. aeruginosa*, and *S. enteritidis* (MIC < 25 μg mL^−1^). They believed that the unique two-dimensional sheet morphology of rhombus lump and rhombus layer CPPs effectively inhibit bacterial growth.

MOFs with two-dimensional morphology exhibit excellent antibacterial properties because their unique morphology can kill bacteria efficiently via physical interaction; however, they can also be harmful to normal cells, posing a risk to human health.

### 3.2. Metal Ion Release

Metal ion release refers to the process of dissociation and release of metal ions from the crystal structure of MOFs in the physiological environment. Various metal ions, such as silver ion (Ag^+)^, zinc ion (Zn^2+^), iron ion (Fe^2/3+^), manganese ion (Mn^2+^), lead ion (Pb^2+^), cobalt ion (Co^2+^), and copper ion (Cu^2+^), have attracted research attention because of their broad-spectrum antibacterial properties and weak toxicity to eukaryotic cells [99,100,101,102,103]. Researchers have synthesized a number of antibacterial MOFs by incorporating the above metal ions into the crystal structure, and they all found that the slow release of metal ions was the determining factor for the good antibacterial properties [53,54,104,105,106]. Liu et al. [50] synthesized three Ag-based MOFs ([(AgL)NO_3_]·2H_2_O, [(AgL)CF_3_SO_3_]·2H_2_O, and [(AgL)ClO_4_]·2H_2_O) with different typologies using tris-(4-pyridylduryl)borane (L), Ag^+^, and different coordination solvents. These three MOFs exhibited a significant bacteria-inhibiting loop with an average diameter of 13–15 mm for *E. coli* and 16−19 mm for *S. aureus*, and MIC values of 250–500 and 293–307 μg·mL^−1^ against the two bacteria, respectively. They attributed the high antibacterial activity to the sustainable and trace release of Ag^+^ (0.85–1.20‰ of overall Ag^+^ in the samples per day). Aguado et al. [49] synthesized two Co-based MOFs (ZIF-67 and Co-SIM-1) using 2-methylimidazole and 4-methyl-5-imidazolecarboxaldehyde, respectively, as the organic ligands. The results showed that the growth inhibition ratio for *S. cerevisiae*, *P. putida*, and *E. coli* was higher than 50% when the concentrations of ZIF-67 and Co-SIM-1 were in the range of 5–10 mg L^−1^. Similarly, they attributed the high antibacterial activity of these MOFs to the sustainable and trace release of Co^2+^.

To further research how released ions kill bacteria in the physiological environment, Taheri et al. [80] explored the degradation process of ZIF-8 in organisms. The experimental data showed that ZIF-8 first released Zn^2+^ and then immediately produced Zn_3_(PO_4_)_2_ (a broad-spectrum antibacterial agent) when immersed in PBS, which simulated the physiological environment (Figure 3B(a,b)). So, they attributed the high antibacterial activity of ZIF-8 to the production of Zn_3_(PO_4_)_2_ in the physiological environment.

Research has shown that MOFs can achieve efficient and long-term antibacterial effects through the slow release of metal ions. However, the specific metabolic behavior of released metal ions in the organism is unclear, and the antibacterial theory of the released metal ions is not unified. Some people believe that the released metal ions directly kill bacteria, while others believe that they form an antibacterial agent with organic substances to kill bacteria [80].

### 3.3. Organic Ligand Release

Organic ligand release refers to the process of dissociation and release of organic ligands from the crystal structure of MOFs in the physiological environment. There are many available organic antibacterial agents, including metallo-organic agents, aldehydes, phenols, acyl aniline, and heterocyclics. These antibacterial agents bind with calcium and magnesium cations of bacterial cells, which leads to the fragmentation of cellular DNA [107]. Fortunately, plenty of these organic antibacterial agents (imidazoles, benzimidazole dazoles, heterocyclic compounds, etc.) can be used in the construction of MOFs [108,109]. Restrepo et al. [52] synthesized a Zn-based MOF ({[Zn(μ-4-HZBA)_2_]_2_·4(H_2_O)}_n_) using 4-HZBA as the organic ligand and Zn^2+^ as the metal center. The experimental data showed that this material significantly inhibited the growth and metabolic activity of *S. aureus*, with a half maximal effective antibacterial concentration of about 20 mg L^–1^, and it released ligands continuously over a period of days. The antibacterial effect of this material was attributed to the release of 4-HZBA; the release of Zn^2+^ not only failed to provide an antibacterial effect, but also promoted the growth and propagation of bacteria to form biofilm. The reason the Zn^2+^ from this material failed to kill bacteria is that it is one of the twenty-six essential elements in animal tissues and its concentration must be above a certain threshold (50 μM) [110] to exhibit an antibacterial effect even though it is a broad-spectrum bactericidal agent.

Research has shown that the release of organic ligands can achieve efficient and long-lasting antibacterial effects. However, the metabolic behavior of MOF-released organic substances in the physiological environment is still unclear, which raises concerns regarding biosafety.

### 3.4. Antibiotic Load

Antibiotics are effective at inhibiting bacterial colonization and growth, and their slow release in the physiological environment can achieve antibacterial purposes. MOFs with high specific surface area are suitable carriers for loading antibiotics, and the released antibiotics can effectively kill bacteria in the physiological environment. Chen et al. [81] synthesized a VAN-functionalized Zr-based MOF (PCN-224) for targeting and killing *S. aureus*. The functionalization endowed the VAN-PCN-224 with excellent antibacterial efficiency against *S. aureus* because of the targeting ability and antibacterial activity of VAN against Gram-positive bacteria. Lin et al. [82] encapsulated VAN into MOF-53(Fe) with strong electrostatic interactions using the one-pot method. This product had an effective drug-carrying capacity of 20 wt% and good biocompatibility. The results showed that this material exhibited a long-lasting and highly effective antibacterial effect (99.3%), because VAN and Fe^3+^ were released slowly in the inflammatory bacterial environment (Figure 3C).

Antibiotics are effective at inhibiting bacterial colonization and growth. However, the abuse of antibiotics has led to the growth of multidrug-resistant pathogens, posing a serious threat to human health.

### 3.5. Gas Load

Gas therapy is a treatment that produces gases to kill bacteria in the physiological environment through degradation or redox reactions. In recent years, some gas molecules, such as H_2_, NO, and CO [111,112,113,114], have been reported to have antibacterial properties and to be effective at alleviating the inflammatory response [115]. MOFs with high specific surface area can be suitable carriers for loading a gas donor, and the released gas from the gas donor can effectively and safely kill bacteria in the physiological environment. GSNO is a donor of NO, which is commonly present in human blood and can decomposed into one NO equivalent and disulfide [116]. Tuttle et al. [83] encapsulated GSNO in a Cu-based MOF (CuBTTri). They found that the decomposition ratio of GSNO in the reaction catalyzed by CuBTTri was only 10% within 16 h without GSH, while in the reaction catalyzed by CuBTTri it was up to 100% with stoichiometric GSH. Therefore, this material can produce more NO and has better bactericidal effect in a bacterial infection environment with higher GSH expression. Wan et al. [84] encapsulated L-Arg as an NO donor into a Zr-based MOF (PCN-224) for gas therapy. After irradiation with a 660 nm laser at 30 mW cm^−2^, L-Arg reacted with H_2_O_2_ to produce NO with a long half-life and wide diffusion range. Guan et al. [85] first synthesized an Hf-based MOF (UIO-66-OH(Hf)) using Hf^4+^ as the metal center and H_2_BDC as the organic ligand. Subsequently, a PS (2I-BODIPY) was attached to UIO-66-OH(Hf) by etherification, and finally the gas donor MnCO was coordinated to the Hf cluster node. They found that the Mn^1+^ in MnCO could oxidize to Mn^2+^ and release CO when the material was exposed to light irradiation (green laser, 0.5 W/cm^2^, 10 min) and oxidants containing H_2_O_2_ and ^1^O_2_. Zhang et al. [86] encapsulated hydrogen-stored Pd nanoparticles into the inner cavity of ZIF-8 by the one-pot method, and then wrapped ascorbyl palmitate hydrogel to target *H. pylori*. The ZIF-8 shell of Pd(H)@ZIF-8 broke and subsequently released high concentrations of Pd nanoparticles and the H_2_ stored in the Pd nanoparticles to kill bacteria in an acidic microenvironment induced by bacteria. It also upregulated the expression of mucosal repair proteins to repair damaged gastric mucosa (Figure 3D).

Research has shown that antibacterial gases are effective at alleviating the inflammatory response and promoting tissue restoration. However, their efficacy in sterilization is limited, so they are generally used as an adjunctive treatment in related investigations.

### 3.6. CDT

ROS are intermediate chemical species formed during the incomplete reduction of oxygen, mainly including H_2_O_2_, ⋅OH, ⋅O^2−^, and ^1^O_2_ [117]. Many natural enzymes can catalyze ROS production from low concentrations of H_2_O_2_, which can kill bacteria by irreversibly damaging bacterial cell walls/cell membranes/cells, DNA, proteins, polysaccharides, and nucleic acids [118]. However, the high cost, complicated purification process, poor stability, and difficult recycling of natural enzymes limit their practical application. It is urgent to find enzyme-like materials that can simulate the activities of these natural enzymes.

Recently, MOF-based catalytic systems mainly based on the simulation of GOD, POD, SOD and CAT were developed, and researchers found that the MOFs could simulate catalytic activity to produce ROS or their precursors for sterilization [119,120,121,122,123]. Zhang et al. [124] first found in 2016 that amorphous Fe NPs can undergo a Fenton-like reaction in the tumor microenvironment of hydrogen peroxide overexpression. The overall Fenton-like reaction involves the ionization of amorphous Fe NPs to release ferrous ions and the subsequent disproportionation of H_2_O_2_ to effectively produce hydroxyl radicals, which will cause irretrievable damage to DNA, lipids, and proteins [125,126]. They also referred to this treatment, which relies on Fenton-like responses, as CDT, and the overall Fenton-like response equation with Fe^2+^ as reactant is as follows:

Fe^2+^ + H_2_O_2_ →Fe^3+^ + (OH)^−^ + ·OH ①

H_2_O_2_ + 2Fe^3+^ → 2Fe^2+^ + O_2_ + 2H^+^ ②

O_2_ + Fe^2+^→ Fe^3+^ + O_2_^−^ ③

Over the past several years, studies of CDT in MOFs have developed rapidly. This is mainly attributed to the intrinsic structural features of MOFs. In an acidic bacterial infection microenvironment, the coordination bonds between metal centers and organic ligands are easily broken, releasing metal ions and thus providing catalysts for CDT. Burachaloo et al. [87] synthesized an Fe-based MOF (MIL-88B). They found that Fe^2+^ was released when MIL-88B was fragmented and subsequently induced an intracellular Fenton-like reaction to produce high concentrations of ·OH. Hao et al. [88] synthesized a Cu-based MOF. They found that Cu^2+^ was reduced to Cu^1+^ as GSH was consumed in the bacterial infection microenvironment, and the Cu^1+^-based MOF catalyzed the formation of ·OH from hydrogen peroxide to kill bacteria.

Research has shown that CDT is a relatively safe treatment because it does not require the introduction of antibiotic drugs, which could cause drug resistance. However, the pH of the bacterial microenvironment does not accommodate the requirements for an effective Fenton reaction (pH = 3.0–4.0) and there is a limited concentration of hydrogen peroxide in the bacterial infection microenvironment, both of which greatly limit the effectiveness of CDT.

### 3.7. PDT

PDT is a treatment that combines PSs and their corresponding light irradiation to catalyze ROS production through photodynamic reactions to selectively destroy target tissues in the presence of oxygen molecules. Photoexcited triplet PSs can react in two ways: (I) directly with the substrate or solvent through hydrogen or electron transfer to form free radicals, or (II) through energy transfer to oxygen molecules to form ^1^O_2_.

PSs can produce ROS to kill bacteria when irradiated by a specific wavelength of light. However, the aggregation between PS nanoparticles leads to the quenching effect. To maintain the ability of PSs to produce photogenerated ROS, researchers have been working on covalently binding PSs, as OBUs, to IBUs and periodically assembling them into porous frameworks to prevent the quenching effect. In addition, the wavelength of light that PSs can absorb is also an important parameter. For biomedical applications, wavelengths are usually controlled in the “tissue transparency” window of 650–800 nm. This is because shorter wavelengths (<600 nm) have limited treatment depth and can cause skin photosensitization, while longer wavelengths (>800 nm) do not provide sufficient energy to produce ^3^O_2_, which is a precursor of ROS [127]. Lu et al. [89] synthesized an Hf-based MOF (DBP-UIO) using Hf^4+^ and the free base porphyrin derivative H_2_DBP under solvothermal conditions. This nanoplate-shaped material had a diameter of 100 nm and a thickness of 10 nm and exhibited photodynamic activity. They believed that the interaction of the heavy Hf^4+^ with the DBP ligand induced an enhanced inter-system crossover from monomorphic to triplet DBP, thus DBP-UIO was at least twice as efficient as H_2_DBP in producing ^1^O_2_ and showed excellent PDT effect. Liu et al. [90] synthesized an Hf-based MOF using Hf^4+^ as the metal center and TCPP as the organic ligand. Under irradiation by an 808 nm laser at 5 mW cm^−2^, this MOF produced more ^1^O_2_ compared to free TCPP. They believed that the coordination of the four carboxyl groups of TCPP with metal ions broadened the Soret band of TCCP and slightly redshifted the four Q-bands. The proximity of heavy Hf^4+^ centers promoted the inter-system crossover of TCPP from the ground state to excited state, and the unique porous structure of the Hf-based MOF led to a good spatial segregation effect between TCPP ligands, avoiding agglomeration and self-quenching of the excited state.

The non-porphyrin photosensitizer BODIPY is also considered to be an ideal PDT reagent because of its low dark toxicity and high extinction coefficient in the therapeutic window. Wang et al. [6] obtained a daughter Zr-based MOF (UIO-PDT) that showed PDT effects by exchanging the I2-BDP ligand with the H_2_BDC ligand of the already synthesized parent MOF via solvent-assisted ligand exchange. The mechanism of the overall solvent-assisted ligand exchange process can be explained as the exposure of the parent MOF to the solution containing a high concentration of the second ligand, then the ligand of the parent MOF being exchanged with the target ligand to obtain a daughter MOF. The experimental results showed that the daughter MOF presented a stronger ability to produce photogenerated ROS and retained the topology of the parent MOF. In addition, the UIO-PDT showed little cytotoxicity to a series of normal cells, in which the cell survival rates were all above 90%, although the concentration of UIO-PDT was adjusted up to 1.0 mg mL^−1^.

PDT is highly selective therapy. It can generate ROS with more efficient bactericidal activity than organic antibacterial agents [128,129,130,131]. However, the production of excessive ROS also causes cellular damage and poses a threat to the health of the organism when cells are affected by microenvironmental stimulation or antioxidant system dysregulation.

### 3.8. PTT

PTT is a treatment in which materials generate heat after absorbing externally stimulated light and kill bacteria by high temperature in the applied environment [132,133]. The principles of photothermal conversion include the following: (I) The electron relaxation of the excited state decays back to the ground state, causing a collision of emitting chromophores with the surrounding environment and partial energy release in the form of heat. (II) Light directs the polarization of free electrons and the depolarization of accumulated charges, leading to collective electron oscillations, surface plasma jump decay, and energy dissipation in the form of heat. (III) When a semiconductor is irradiated with high-energy light, the electrons are excited to higher energy levels in the conduction band and leave holes in the VB. The electrons and holes will relax to the edge of the energy band by vibrational relaxation, which leads to the conversion of energy into heat. MOFs have attracted much attention in recent years in the field of photothermal antibacterial therapy because of their semiconductor-like behavior. In addition, some MOFs can also generate heat to kill bacteria under light irradiation because of the strong LSPR of metal ions at the active sites.

PB presents a face-centered cubic structure formed by the coordination of a carbon atom and a nitrogen atom from a carbon–nitrogen triple bond with Fe^2+^ and Fe^3+^, respectively. PB and its analogs comprise the most important branch of intrinsic photothermal MOFs. It is also one of the oldest synthetic MOFs and widely used as a PTA, which has been approved by the Food and Drug Administration as a clinical drug for light radiation therapy [134]. Fu et al. [92] synthesized PB with controlled size by mixing solutions of FeCl_3_ and K_4_[Fe(CN)_6_] with citric acid as the surface capping agent. The experimental data showed that the PB had a broad absorption band at 500–900 nm with an absorption peak at 712 nm. Moreover, under irradiation by an 808 nm laser at 0.6 W cm^−2^, the PB had a molar extinction coefficient of 1.09 × 10^9^ M^−1^ cm^−1^, which is slightly lower compared to Au nanorods, and its temperature increased to 43 °C in less than 3 min. However, the absorption peak was located at the edge of the near-infrared region (700–900 nm), and the photothermal conversion efficiency was only 20% at 808 nm irradiation [135].

Other intrinsic photothermal MOFs are based on the LMCT mechanism to achieve an antibacterial effect, which tends to occur when organic ligands and metal ions are in relatively low valence states. IR825 has an absorption peak at 825 nm in the near-infrared light region and is commonly used as near-infrared dye. Yang et al. [93] synthesized an Mn-based MOF using Mn^2+^ as the metal center and IR825 as the organic ligand. The temperature of this material increased rapidly to approximately 52 °C within 5 min under irradiation by an 808 nm laser at 0.6 W/cm^2^, and it had a mass extinction coefficient of 78.2 L g^−1^ cm^−1^ and good photothermal cycling capability. Deng et al. [94] used Fc(COOH)_2_, a widely used PTA, as the ligand and Zr^4+^ as the metal center to synthesize a nano-sheeted Zr-Fc MOF. The Zr-Fc MOF had broad light absorption in the range of 350–1350 nm, and its temperature reached 92 °C within 3 min under irradiation by an 808 nm laser at 0.2 W cm^−2^, which was significantly higher than Fc(COOH)_2_ (around 46.8 °C). The higher photothermal conversion efficiency of Zr-Fc MOF stemmed from its higher stability, so most excited photoelectrons decayed through non-radiative pathways and consequently generate heat. Liu et al. [95] synthesized an Fe-based MOF (Fe-CPND) using Fe^3+^ as the metal center and GA as the organic ligand. In an acidic environment, the GA ligand gradually dissociated and the Fe-CPND transformed from triple to double ligand, conferring a higher longitudinal relaxation rate. They also found that the preferable photothermal effect of Fe-CPND resulted from the LMCT effect triggered by the Fe–phenol structure.

The high temperature induced by PTT can kill most bacteria that are intolerant to high temperatures. However, when the temperature is too high and insufficient to completely remove the bacteria, non-local heating usually causes severe damage to healthy tissues and an inflammatory response [136].

### 3.9. SDT

Because of the shallow tissue penetration depth of PDT and the damage to normal cells caused by long-term exposure to external stimulating light, its clinical application is limited. SDT is a treatment in which the sound sensitizer reacts with oxygen to produce ROS to kill bacteria under ultrasonic radiation. The cavitation effect occurs during the ultrasound process, and the resulting bubbles rapidly compress in volume and burst to produce strong shock waves, local high temperature, high pressure, and hydroxyl radicals to improve the antibacterial effect. Pan et al. [96] synthesized PMCS containing porphyrin-like metal centers. PMCS can be obtained after calcination at high temperature using ZIF-8 as a template, and it is a kind of N-doped carbon material with a Zn-centered porphyrin-like structure. The unique micro/mesoporous structure of PMCS enhances the cavitation effect, resulting in high SDT efficiency. Experimental results showed that the ^1^O_2_ producing efficiency of PMCS was increased by 203.6% and it could generate more ·OH than ZIF-8.

Pan et al. [97] synthesized an MOF-derived DHMS nanoparticles. DHMS can be excited by US irradiation to produce electrons and holes, which interact with O_2_ and H_2_O in the body to produce ^1^O_2_ and ·OH, respectively. The low-valent Mn in DHMS is partially oxidized to high valence by the holes during the sonication reaction, promoting electron–hole separation and ROS production. In addition, the hollow pore structure of DHMS enhances the cavitation effect to further increase ROS generation. Yu et al. [98] synthesized a US-activated single-atom catalyst (HNTM-Pt@Au) actuated by Au nanorods. HNTM is a Zr-based MOF with a porphyrin-like structure and works as a carrier in this material. The experimental data showed that the antibacterial efficiency of HNTM-Pt@Au against methicillin-resistant *MRSA* under US was 99.93%. They summarized the ultrasonic catalysis mechanism of this material as follows: (I) Au NRs enhance ultrasonic cavitation and improve the absorption of ultrasonic energy in the system, and (II) Au NRs and single Pt atoms act as electron acceptors to promote the electron transfer generated by HNTM and improve the separation efficiency of electron–hole pairs. In addition, the products obtained by the exchange of single Au and Cu atoms by single Pt atoms in HNTM also generate a large amount of ROS, which can be attributed to the metal with a higher work function having a higher carrier separation efficiency.

Intermittent low-power ultrasound is harmless to the human body, so this non-invasive therapy has a bright future in clinical applications. However, if the ultrasound treatment time is too long or the heat is not controlled properly, it can lead to burns, inflammation, or even necrosis in the skin or deep soft tissues.

### 3.10. Synergistic Therapy

Taking advantage of the synergistic effect of multiple therapies is an effective way to improve the antibacterial activity. In recent years, many MOFs have achieved the effect of “1 + 1 > 2” by synergizing the antibacterial activity of their components. To synthesize MOF-derived 2D carbon nanosheets, Fan et al. [137] first synthesized MOF-derived ZnO on graphene, and then obtained TRB-ZnO@G by in situ polymerization of anchored phase change TRBs. The TRB-ZnO@G not only had nearly 100% antibacterial efficiency at low concentrations, but also exhibited rapid and safe skin wound disinfection, without damaging normal skin tissue or causing cumulative toxicity. Under NIR irradiation, TRBs absorbed NIR light and increased the local temperature, which also promoted the release of Zn^2+^ from ZnO and penetrated into bacterial cells. Moreover, the unique two-dimensional structure of 2D carbon nanosheets enhanced the physical interaction with bacterial cells. Thus, the antibacterial mechanisms working synergistically exhibited more effective antibacterial activity than any single mechanism component.

Synergistic therapy is not simply the sum of individual antibacterial mechanisms but involves the complementarity of the strengths and weaknesses of various mechanisms for mutual improvement to achieve better antibacterial effects. The synergistic antibacterial activity of MOF-based composites will be discussed in more detail in the next section.

## 4. Strategies to Enhance the Antibacterial Ability of MOFs

Although MOFs have shown good antibacterial effect in the experimental stage, their antibacterial ability has not yet reached clinical demand because of the complexity of the human body microenvironment. Therefore, researchers have launched extensive investigations into strategies for enhancing the antibacterial properties of MOFs in recent years. They found that these properties can be improved by modulating the size, pore size, and coordination environment of active sites of MOFs, and by synthesizing MOF-based composites. These strategies are discussed in detail in this section.

### 4.1. Size Modulation

The size effect influences the uptake of antibacterial MOFs by bacterial cells. Bacterial cells tend to uptake nanoparticles of small size, so the uptake of MOFs by bacterial cells can be improved by reasonably regulating the size of MOFs (Figure 4A). In addition, miniaturization could increase the interactions between MOFs and bacteria [138,139], which would enhance their antibacterial ability. Akbarzadeh et al. [55] synthesized a nanoscale Zn-based MOF (Zn-PDA) using Zn(NO_3_)_2_ and PDA. The nanoscale Zn-PDA exhibited better antibacterial properties against *S. aureus*, *S. enteritidis*, *A. baumannii*, *K. pneumoniae*, *S. entica*, and *E. coli* (average inhibition diameter, 8.6–17 mm; MIC, 300–308 μg mL^−1^) compared with the large size Zn-PDA. They believed that the small particle size and high surface area gave the nanoscale Zn-PDA good antibacterial effect.

Recent studies have shown that nanoscale MOFs are commonly used for antibacterial application because their miniaturized size corresponds to a high specific surface area, which increases their interaction with bacteria and promotes their penetration into bacterial cell membranes. In bacterial cells, nanoscalable MOFs can interact with lipophilic acid, phosphate, or hydroxyl groups, leading to cell destruction [101,140,141] and greatly increasing the antibacterial effect. However, the strong permeability of cells to nanoscale MOFs also means that they can penetrate cell membranes and accumulate in normal cells, which would a pose risk to human health.

### 4.2. Pore Size Modulation

Rational design of MOF pore size can facilitate their use as carriers to load drug molecules (gas donors, antibiotics, etc.) for sterilization. However, drug molecules vary in size, and the pore size of the MOF should match the size of the drug molecule to ensure that the drug can be successfully loaded into the pores (Figure 4B). Pore size modulation is a strategy for obtaining MOFs with different pore sizes by changing the synthesis method, reactant ratio, and post-treatment method. Zhang et al. [142] synthesized structurally controlled ZIF-8 with a 3D mesoporous structure using polystyrene as sacrificial template. ZIF-8 precursor solution was impregnated into polystyrene particles to synthesize ZIF-8, and then polystyrene was dissolved to obtain ZIF-8 with a porous structure, with the original position of polystyrene particles becoming the pores. They believed that modulating the pore size by regulating the size of PS particles is a suitable method for designing MOFs with pore sizes matched to drug molecule sizes.

Xing et al. [143] synthesized a flower-like ZIF-8 by adding dopamine as an additional ligand to reduce the molar ratio of Hmim/Zn^2+^ during the synthesis process. The ligand-polymerized molecular sieve imidazole framework catalyzed the in situ polymerization of polydopamine, leading to microporous blocking and cross-linking of morphologically degraded ZIF nanosheets. Under the protection of polydopamine, flower-like ZIF-8, with abundant micropores, mesoporous defects, large petal spacing, high specific surface area, and high metal atom loading, was obtained by carbonization. They found that the flower-like ZIF-8 showed excellent POD activity and produced more effective ROS than normal ZIF-8 because of its wider 3D accessibility of active sites.

Yang et al. [144] synthesized mesoporous polymeric carbon nitride (PCN) and used ascorbic acid-assisted hydrothermal etching of bulk PCN to form pores and improve its crystallinity. PCN has many lone pairs of electrons on its N atoms. When PCN is placed in an inorganic weak acid solution (ascorbic acid), the acid attacks the lone pairs of electrons and the Lewis acid–base reaction occurs [145], corroding the PCN and forming pores. This simple synthesis step expands its specific surface area and provides transfer channels, which can potentially promote its photocatalytic hydrogen production (26.8 μmol h^−1^). Deng et al. [146] synthesized ZIF-8-encapsulated Au nanoflowers (ZIF-8@Au nanoflowers), and then selectively etched them with tannic acid into a material with yolk-shell structure (Figure 5A(a)). The results showed that ZIF-8 became a very thin shell layer, and more cavities appeared between it and the Au nanoflowers, which had greater drug loading ability compared to the ZIF-8@Au nanoflowers with core–shell structure (Figure 5A(b)). Moreover, the core Au nanoflowers still maintain good photothermal effect (Figure 5A(c)).

Studies have shown that porous materials with adjustable pore size not only are helpful for drug loading, but also could have high specific surface area via structural changes. However, other functions of porous MOFs, such as stability, adsorption dynamics, processability, and mechanical and thermal properties, which are essential for future practical applications of the materials, have been less studied.

### 4.3. Modulating Coordination Environment of Active Sites

Active sites are where chemical reactions occur in materials, and the coordination environment refers to the specific binding mode between ions at active sites. A metal center connected to ions provided by organic ligands constitutes the coordination environment of MOFs [148]. The ions in the coordination environment are slowly released in the physiological environment and express their special properties. Rational modulation of the metal centers in the coordination environment could re-endow MOFs with different antibacterial mechanisms. Moreover, rational modulation of the ions provided by organic ligands in the coordination environment leads to local charge inhomogeneity at the active sites, which could promote carrier separation and ultimately enhance the photodynamic effect of the material. So, modulating the coordination environment of active sites is a strategy for designing MOFs with desirable physicochemical properties by changing their metal centers and the ions provided by organic ligands (Figure 4C).

In recent years, MOFs and derivatives with altered coordination environments have attracted significant attention because of their near 100% metal dispersion [149,150,151]. However, methods of synthesizing MOF derivatives with same coordination environment vary slightly; some researchers use the one-pot method, in which a solution containing secondary metal ions or complexes composed of secondary metal ions and organic substances is added directly to the template precursor solution to synthesize MOFs with multiple types of active sites, then the desired coordination environment is retained, and unwanted ones are removed by calcination. Han et al. [152] loaded Fe-Phen complexes into ZIF-8 precursor solution via the one-pot method, and then pyrolyzed them under argon atmosphere. Finally, they synthesized a well-dispersed single-atom catalyst with good electrocatalytic activity. Cao et al. [153] synthesized 20 single-atom enzymes with different metal-N coordination environments (metal = V, Cr, Mn, Fe, Co, Ni, Cu, Ce, Zr, Mo, Ru, Rh, Pd, Gd, W, Re, Ir, Pt, Au, Tb). They added metal salt solution containing secondary metal ions in proportion to the primary metal ion (Zn^2+^) and a solubilizer with antibacterial properties to the precursor solution of ZIF-8 and obtained products by sintering and centrifugation. The experimental data showed that these materials all had the catalytic behavior of OXD, POD, and simulated HPO; the Fe-based MOF with Fe-N coordination environment exhibited the highest POD activity. In addition, they found through DFT that the catalytic pathway and reactivity of ROS were closely related to the electronic structure of the metal centers, and higher ROS catalytic properties could be guaranteed if the energy barrier for ROS generation was low.

In some studies, researchers adsorbed complexes composed of secondary metal ions and organic substances onto the surface and into pores of nitrogen–carbon obtained by calcination using MOFs as templates, and then formed a new coordination environment after calcination. Wang et al. [154] synthesized a Cu-based MOF (Cu SASs/NPC). They removed the Zn^2+^ from ZIF-8 by high temperature and obtained nitrogen–carbon via acid etching, and then dipped it into a complex composed of Cu^2+^ and dicyandiamide. Finally, the complex was adsorbed on the original active sites of Zn-N_4_. When dicyandiamide was burned off by high temperature calcination, the encapsulated Cu^2+^ was exposed and combined with four nitrogen atoms to form a new Cu-N_4_ coordination environment. The experimental data show that Cu SASs/NPC had not only the peroxidase-like activity of ZIF-8, but also the photothermal properties of Cu^2+^. The photothermal conversion efficiency of Cu SASs/NPC was calculated to be 82.78%, which was much stronger than the previously reported Cu-based photothermal agents (<50%), and the survival cell ratio of the Cu SASs/NPC + H_2_O_2_ + NIR light group against *E. coli* and *MRSA* reached 0%.

Other researchers found that co-calcining template MOFs directly with metal powder or foam also could form MOFs with new coordination environments [155]. Moreover, changing the ions provided by organic ligands in the coordination environment of active sites to ions with different electronegativity will cause uneven local charge distribution, which can accelerate the current transfer and catalyze the redox reaction of the material, producing antibacterial substances. Ji et al. [147] found that replacing the N ion in ZIF-8 with low electronegativity ions was a way to adjust the electrocatalytic properties of materials. They synthesized a heterojunction of Co-based MOF and CdS (Co-N_X_PS/C/CdS), in which P ions were introduced by adding triphenylphosphine during the MOF synthesis process, and S ion was introduced by the co-calcination of MOF and sulfur powder (Figure 5B(a,b)). The experimental data showed that the coordination environment of M-N_X_PS had an inhomogeneous charge density compared to M-N_4_. The inhomogeneous charge density can enhance light absorption as well as collect and store photoexcited electrons from CdS. Therefore, promoting charge carrier separation to a large extent and boosting photocatalytic hydrogen production (Figure 5B(c)).

In addition, the effects of high temperature on the structure and properties of MOFs have been studied extensively recently. It was found that high temperature does not significantly affect the morphology of MOFs, but the physical and chemical properties are slightly different. Wang et al. [156] explored changes in the crystal structure of ZIF-8 at various temperatures (500–900 °C). At temperatures less than 600 °C, tetragonal planar zinc porphyrin-like centers are formed on the nitrogen–carbon substrate, and the Zn^2+^ tends to move to the N_4_ sites in the nitrogen–carbon substrate plane as the pyrolysis temperature increases. ZIF-8 synthesized at 800 °C had the best bond strength at Zn-N_4_ active sites, which can maximize the adsorption of H_2_O_2_ and exhibit excellent peroxidase-like activity.

Studies have shown that series of MOFs with desirable properties can be designed by modulating the coordination environment of active sites, and this strategy has been widely used in recent years for the preparation of novel MOF derivatives. However, the high temperature will also change the carbon structure, bond length, and bond angle of the original MOF, causing problems when making side-by-side comparisons.

### 4.4. Constructing MOF-Based Composites

In MOF-based composites, the antibacterial effect can be enhanced, and the antibacterial mechanisms of individual components can be synergized, so they have better antibacterial ability than each component would have. In recent years, the synthesis of composites using MOFs with other components has also been a popular research topic (Table 3). The methods of preparing such composites can be divided into those that use MOFs or MOF precursor solution commingled with other components. MOF-based composites can be classified as MOF@metal and the oxidation products MOF@carbon, MOF@MOF, targeted MOF, and stimulus-responsive MOF.

#### 4.4.1. MOF@metal and Oxidation Products

MOF@metal and oxidation products can be obtained by introducing metal ions, metal nanoparticles, and metal oxides into MOFs. Introducing metal ions can change the regularity of the MOF structure and couple the properties of the metal ions, which is expected to improve the antibacterial properties of the composites. Han et al. [56] synthesized a Cu^2+^-doped Zr-based MOF (PCN-224). Under irradiation by a 660 nm laser at 0.4 W cm^−2^, the proper amount of Cu^2+^ doped in PCN-224 can trap electrons, accelerate carrier transfer, suppress electron–hole recombination, and finally catalyze ROS production. In addition, Cu^2+^-doped PCN-224 can convert light energy into heat because of the presence of Cu^2+^, and both of them contribute to antibacterial activity. The experimental data showed that product doped with 10% Cu^2+^ in PCN-224 had the best antibacterial efficacy (99.71%) against *S. aureus* under irradiation by 660 nm laser at 0.4 W cm^−2^ within 20 min. However, as the amount of Cu^2+^ introduced increased excessively, structure regularization effects occurred, ROS production gradually decreased, and the bactericidal effect decreased. Zhang et al. [161] constructed MOFs and mixed-metal MOFs on Ti surfaces through the co-assembly of Au_25_(MHA)_18_ and metals centers (Ti^4+^, Zr^4+^, Hf^4+^, and Cu^2+^). Results showed that the antibacterial efficiency of the Ti/Zr/Hf-based MOFs coating against both *MRSA* and *E. coli* were less than 5%. However, the antibacterial efficiency of Ti/Zr/Hf-Cu mixed-metal MOFs coating against *MRSA* was 98.86 ± 2.53%, 98.66 ± 2.0% and 98.59 ± 1.84% when Cu^2+^ was introduced in twice the amount of M^4+^, respectively. They believed that the doping of boundary acids with Cu^2+^ results in the formation of unstable submetallic-oxygen bonds, which generally promote the release of bactericidal Cu^2+^, leading to bacterial death.

Some metal nanoparticles can be used for antibacterial applications by themselves. However, they are thermodynamically unstable and prone to aggregation; therefore, loss of magnetic, catalytic, or rotational activity can occur [162,163]. The use of MOFs can avoid this limitation because their homogeneous pore structure can stabilize NPs and prevent their aggregation by providing spatial constraints [164,165,166]. Deng et al. [57] loaded Au NPs onto the ZIF-8 surface and found that the product (Au@ZIF-8) was a heterojunction of two semiconductors (Figure 6A(a,b)), which formed a Schottky junction at the Au and ZIF-8 interface, increasing the electron density of AuNPs [167] and contributing to the conversion of ^3^O_2_ to ^1^O_2_ (Figure 6A(c)). Moreover, AuNPs have a strong LSPR at 530 nm, increasing the visible light absorption of Au@ZIF-8. Both of them can enhance the electron transport and charge carrier separation of Au@ZIF-8 (Figure 6A(d)) The experimental data showed that the amount of ·OH produced by Au@ZIF-8 was nine times higher than that produced by ZIF-8, and the inhibition ratio against *S. aureus* and *E. coli* was >99.9% when the dosage of Au@ZIF-8 was 0.2 mg mL^−1^. Deng et al. [146] synthesized star-shaped Au nanoflowers, and then obtained Au nanoflower@ZIF-8 by encapsulating them with ZIF-8. They found that the photothermal properties of Au nanoflower@ZIF-8 under 1064 nm (NIR-II biological window) laser irradiation were not inferior to those under 808 nm (NIR-I biological window) laser irradiation. Therefore, it can be applied to deeper tissues for antibacterial treatment because it can absorb higher wavelengths of light. Yang et al. [58] synthesized a product (SPZA) by loading ZIF-8@Ag onto sulfonated polyetheretherketone. The results showed that the sulfonated polyetheretherketone had a porous and loose structure, which allowed more ZIF-8 and Ag^+^ to be loaded; the number of *E. coli* and *S. aureus* on SPZA was zero because ZIF-8 and Ag^+^ were released synergistically against bacteria.

In recent years, researchers have carried out extensive studies on further inhibiting electron and hole recombination to improve PDT effectiveness. They have found that surface oxygen vacancy defects that form on the surface of oxidized MOF composites can enhance carrier transfer and effectively inhibit electron and hole recombination. Liang et al. [59] synthesized nanorods of MnO_2_/ZIF-8 using the one-step method. The experimental data showed that the MnO_2_/ZIF-8 hybrids exhibited complete inactivation against *E. coli* at low concentrations (3.24 μg mL^−1^). They believed that the antibacterial mechanism of the MnO_2_/ZIF-8 hybrids was mainly attributed to the production of catalytic ^1^O_2_ rather than the release of Zn^2+^ or the photothermal effects under simulated solar irradiation; the increase in surface surface oxygen vacancy defects and the presence of heterojunctions could inhibit the recombination of electrons and holes to catalyze the photogeneration of ROS to kill bacteria. Cui et al. [60] synthesized Ag/ZnO hybrid cages with well-preserved polyhedral shapes and rich mesoporous structures by in situ pyrolysis of the product, obtaining by ZIF-8 impregnated with AgNO_3_. The experimental data showed that the MIC of Ag/ZnO hybrid cages against bacteria was 6.25 μg mL^−1^, which is comparable with commercial medicines such as streptomycin and ciprofloxacin. They believed that the LSPR effect of the encapsulated Ag nanoparticles and the band gap reduction effect caused by slight doping of Ag enhanced the absorption of visible light, resulting in a large number of light carriers. Photogenerated electrons can accumulate on Ag nanoparticles and later lead to rapid charge separation through a Schottky junction at the Ag/ZnO interface because of the high work function of the metal particles. Moreover, the electrons can react with dissolved O_2_ to form ·O_2_^−^ in such semiconductor heterojunction systems, and the holes at the edge of the CB can participate directly in the degradation of oxide species as well as react with H_2_O to form ·OH radicals for sterilization (Figure 6B). Wang et al. [136] synthesized a Z-type heterojunction of g-C_3_N_4_ and Cu_3_P (g-C_3_N_4_/Cu_3_P), which improved the separation efficiency of photogenerated electrons and holes, resulting in high production of H_2_ and ROS. This process took advantage of the low acidity, high GSH content, and high H_2_O_2_ content of the bacterial infection microenvironment to achieve H_2_-mediated cascade amplification of synergistic hydrogen therapy/PTT/PDT/CDT.

Research has shown that MOF@metal and oxidation products can prevent the aggregation quenching effect between metal nanoparticles. In addition, these MOF-based composites will obtain stronger antibacterial ability than individual components by the synergy of the antibacterial mechanisms of MOFs themselves and the introduced metal ions. However, if too many metal ions are introduced, it will inevitably cause metabolic toxicity in the physiological environment.

#### 4.4.2. MOF@carbon

Carbon-based materials include 0-dimensional CQDs and GQDs, one-dimensional CNTs, two-dimensional graphene and its derivatives, such as GO and RGO, and three-dimensional AC. They are well suited for antibacterial applications because of their unique physical interaction ability, photoelectric capability, and biocompatibility [168,169,170,171]. However, carbon-based nanomaterials, especially small-sized carbon nanoparticles, are prone to clustering and quenching, greatly reducing the effectiveness of their antibacterial properties. In recent years, many researchers have compounded carbon-based materials with MOFs. The porous MOFs can hinder the clustering and quenching effect of carbon-based materials and improve the stability and antibacterial effect of MOF@carbon. Travlou et al. [157] synthesized an Ag-based MOF@carbon composite antibacterial agent (Ag-BTC-S-/N-CQDs). This composite had superior bacterial inhibition ability compared to its individual components (S-/N-CQD or Ag-BTC) because of the synergistic effect of Ag^+^ antibacterial release and the physical interaction of CQD with bacterial cells. Hatamie et al. [61] synthesized a Co-based MOF@carbon composite antibacterial agent (GO/Co-PTA). The experimental data showed that it inhibited the growth of *E. coli* and *S. aureus* by up to 99%, which was attributed to the synergistic antibacterial effects of Co^2+^ release and the physical interaction of GO with bacterial cells (Figure 6C).

Research has shown that MOF@carbon can prevent the aggregation quenching effect between carbon nanomaterials through the spatial segregation effect. In addition, the synergistic effects of MOFs and carbon materials can further improve the antibacterial activity of composites. However, carbon nanomaterials are expensive and cannot be put into mass production for clinical use.

#### 4.4.3. MOF@MOF

MOF@MOF mostly refers to dual MOFs with core–shell structure, in which both the core and shell can be used as carriers to load antibacterial substances or can themselves be considered as antibacterial agents. In recent years, core and shell MOFs synthesized from PS-type and PTA-type MOFs have been extensively studied, and many MOF-based composites have been obtained with both PDT and PTT effects [172,173]. Liu et al. [158] synthesized a dual MIL-101 with core–shell structure, and then obtained a drug-loaded MOF (BQ-MIL@cat-fMIL) and used an in situ growth method to load (BQs in nuclear MIL-101 and peroxidase in shell MIL-101. They found that BQs in core MIL-101 had both photodynamic and photothermal activities because of their quantum confinement and edge effects; the photosensitivity of MIL-101 led to a widening of the gap between the triplet and ground states of the composite, which resulted in the light-excited multi-MOF reacting mainly with ^3^O_2_ to produce ^1^O_2_. In addition, the peroxidase in shell MIL-101 provided O_2_ to the internal BQs by decomposing H_2_O_2_ and improved the PDT effect of the BQs (Figure 6E). The experimental data showed that the percentage of hypoxic cell apoptosis was 52.1% when this material was under irradiation by a 660 nm laser at 150 mW cm^−2^, and the photothermal conversion efficiency of BQ-MIL@cat-fMIL was 23.3% under irradiation by an 808 nm laser at 1 W cm^−2^, which induced a 28.7% apoptosis ratio in hypoxic cells; a 75.6% apoptosis ratio in BQ-MIL@cat-fMIL-treated hypoxic cells was achieved after applying dual light irradiation. Luo et al. [62] synthesized dual MOFs with core–shell structure using PB as the core and TCPP-doped UIO-66 as the shell (Figure 6D(a)). PB acted as the PTA and TCPP as the PS, which gave these dual MOFs the effects of PDT and PTT (Figure 6D(b)). Moreover, PB- and TCPP-doped UIO-66 are n-type semiconductors and can recombine to form an n-n heterojunction. In this heterojunction, the CB of PB is lower than that of TCPP-doped UIO-66, therefore these dual MOFs can accelerate photoelectron transfer from PB to UIO-66 and inhibit photoelectron–hole recombination under 660 nm laser irradiation (Figure 5D(c)). The experimental data showed that these dual MOFs had a high yield of ^1^O_2_ under 660 nm light irradiation and high photothermal conversion efficiency up to 29.9% under 808 nm light irradiation; the antibacterial efficiency against *S. aureus* and *E. coli* exceeded 99.31 and 98.68%, respectively. In addition, the dual MOFs released trace amounts of Fe and Zr ions during the degradation process, which is beneficial to wound healing.

Research has shown that both MOFs in MOF@MOF can be used as carriers to load ideal PSs and PTAs to prevent the aggregation quenching effect, or can themselves work as PSs and PTAs to improve the antibacterial effects of the composites. In addition, the specific time and space for the release of different drug molecules in the physiological environment could be regulated by the rational design of core and shell MOFs and loaded drug molecules. However, the process of synthesizing such materials is tedious and costly.

#### 4.4.4. Targeted and Stimulus-Responsive MOFs

Targeted and stimulus-responsive MOFs are MOFs with targeting and stimulus-responsive outer cladding, respectively. Because the antibacterial activity of PDT materials is limited by the ultra-short diffusion distance of ROS, the use of stimulus-responsive or targeted substances to modify materials to achieve precise controlled release of drugs at the correct time and place has a been popular research topic in recent years. Chen et al. [159] integrated BA ligand, which can specifically bind to bacteria and photosensitized porphyrin, into Zr- and Cu-based MOF (multivariate MOF). The results showed that the BA ligand targeted bacteria and the photosensitized porphyrin produced ROS under visible light irradiation, which synergistically improved the antibacterial effect against *MRSA* and *MDR E. coli* (10–20 times higher than without the targeting ligand) and accelerated the healing of chronic wounds infected with *MDR* bacteria (nearly 2 times faster than without the targeting ligand). Ahmad et al. [63] obtained a composite containing ZIF-8, GO, and amino groups (ZGO-NH) by grafting amino groups onto the surface of ZIF-8. The experimental data showed that when the volume of ammonium hydroxide (provider of amino groups) introduced into ZIF-8 was up to 20 mL, the modified ZIF-8 had better antibacterial effect against *E. coli* and *S. aureus* (inhibition zone diameters of 2.59 and 3.82 cm, respectively). They believed that the positively charged protonated amine groups enhanced the interaction between ZIF-8 and negatively charged bacterial cells, and they exhibited better antibacterial effect. Song et al. [64] synthesized a Zn-based MOF with an UV light-responsive rifampicin (RFP) release system (ZIF-8@RFP) by surface modification of ZIF-8 with 2-nitrobenzaldehyde (o-NBA), a pH jump reagent, then doped the RFP into the ZIF-8. The o-NBA decomposed in situ under UV irradiation at 5.25 mW cm^−2^ to produce acids and induce decreased pH in the physiological environment; ZIF-8 was degraded in the acidic environment and released Zn^2+^ and RFP continuously, which may help kill bacteria. The results showed that ZIF-8@RFP exhibited excellent antibacterial properties against MRSA under UV irradiation at 5.25 mW cm^−2^ (antibacterial ratio > 99%) at a dose of up to 10 µg mL^−1^.

Research has shown that targeted and stimulus-responsive MOFs can precisely locate bacteria in space and exhibit stronger antibacterial ability than MOFs without the targeting and stimulus-response functions. However, most targeting and stimulus-responsive materials are harmful to organisms, which limits their clinical application.

## 5. Conclusions and Prospects

Over the past few years, MOFs and MOF-based composites have had a significant impact in the field of biomedicine because of their high specific surface area, atomic utilization, photodynamic activity, and photothermal activity. Moreover, they can control/stimulate the release of antibacterial components and regulate their size and morphology. Although MOFs have many advantages and are candidates for biomedical applications, they are only in the preliminary research stage, and there are still several issues and challenges that need to be addressed.

MOFs are made of metal ions and organic ligands, and it is undeniable that some of them can degrade and produce those components in certain environments. However, there is still a lack of studies on the effects of MOF degradation products on body metabolism. To address safety concerns, pathological research should be carried out. In addition, magnetic thermal therapy, microwave thermal therapy, and other treatment methods that cause less harm to normal cells, which are less studied in the field of antimicrobial therapy, may become a new development direction in the future.

Most MOFs with photodynamic and photothermal activity can only absorb light in the visible and NIR-I region, which limits their application because of the shallow depth of tissue transmission. For better effectiveness, there is an urgent need to find new MOFs with deeper tissue transmission that can absorb longer wavelengths of light.

In addition, researchers have synthesized the same MOF-derived single-atom enzymes with different amounts of active sites using different preparation methods and have found that these materials have excellent antibacterial effects, but no one has yet compared the antibacterial effects of the same single-atom enzymes with different numbers of active sites. Exploring the optimal range of the number of active sites in a material could help enrich the knowledge of antibacterial mechanisms regarding active sites and find the best balance between effectiveness and safety in MOF active sites.

## Figures and Tables

**Figure 1 jfb-13-00215-f001:**
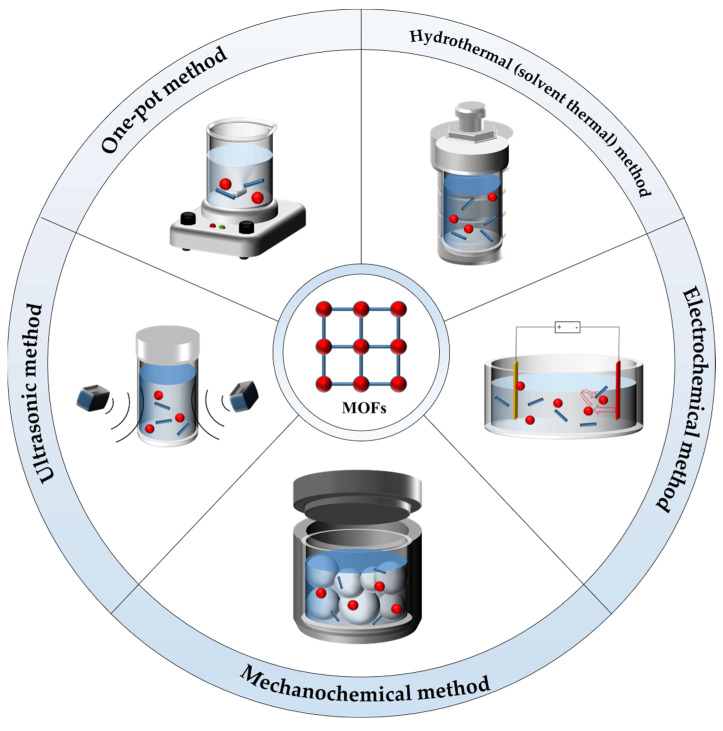
Schematic illustration of methods for preparing MOFs.

**Figure 2 jfb-13-00215-f002:**
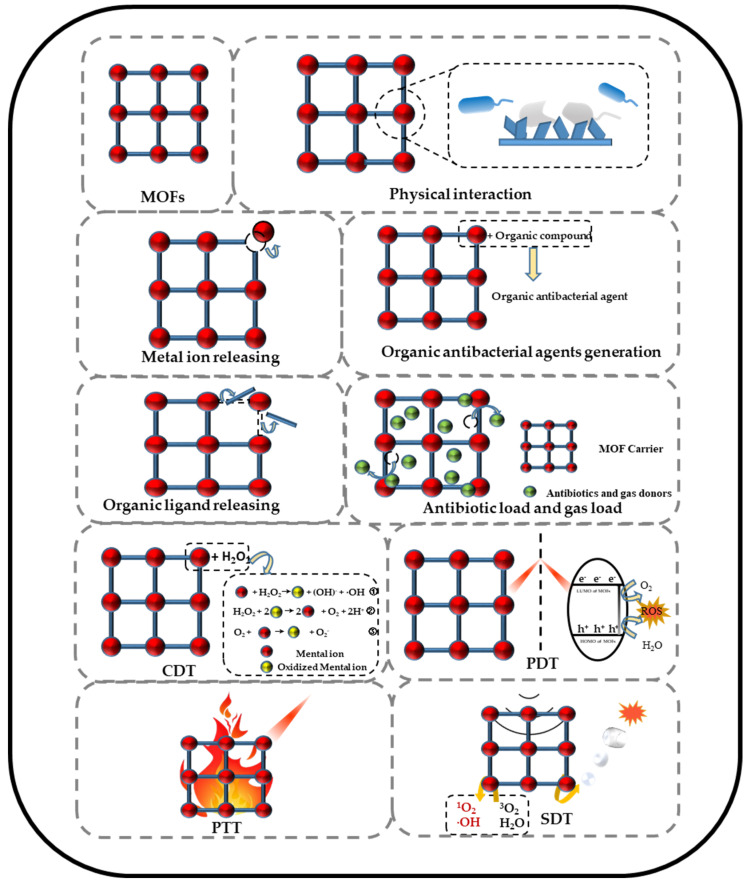
Schematic illustration of antibacterial mechanisms of MOFs.

**Figure 3 jfb-13-00215-f003:**
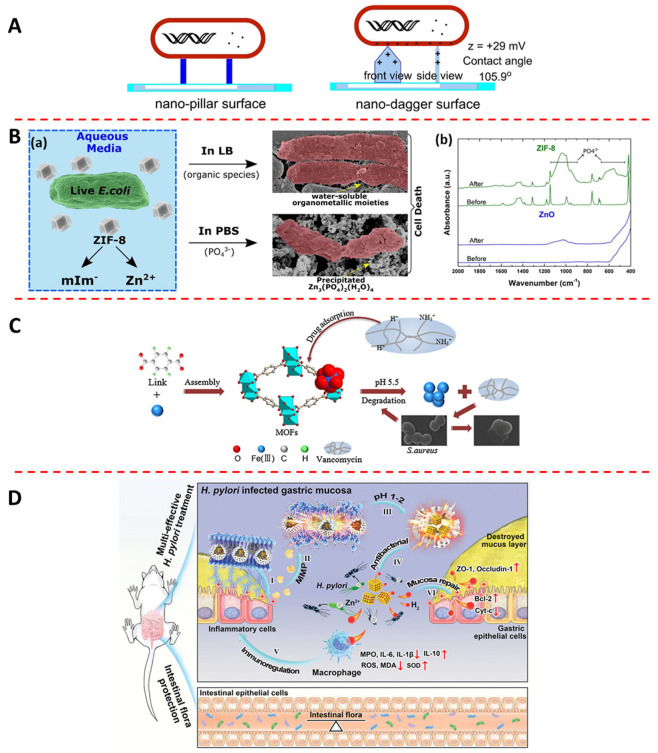
Various antibacterial mechanisms of MOFs: (**A**) Schematic illustration of ZIF−L with nano-dagger morphology in physical interaction with bacteria (adapted from [53]). (**B**) (**a**) Schematic illustration of ZIF−8 reacts with PBS to produce Zn_3_(PO_4_)·2(H_2_O)_4_ with antibacterial activity (adapted from [80]); (**b**) FTIR spectrum of the ZIF−8 and ZnO nanopowders before and after PBS incubation (adapted from [80]). (**C**) Schematic illustration of VAN@MOF−53(Fe) release Fe^3+^ and VAN to kill *S. aureus* in an acidic environment (adapted from [82]). (**D**) Schematic illustration of Pd(H)@ZIF−8 release Zn^2+^, Pd nanoparticles, and H_2_ in an acidic environment to kill H. pylori (adapted from [86]).

**Figure 4 jfb-13-00215-f004:**
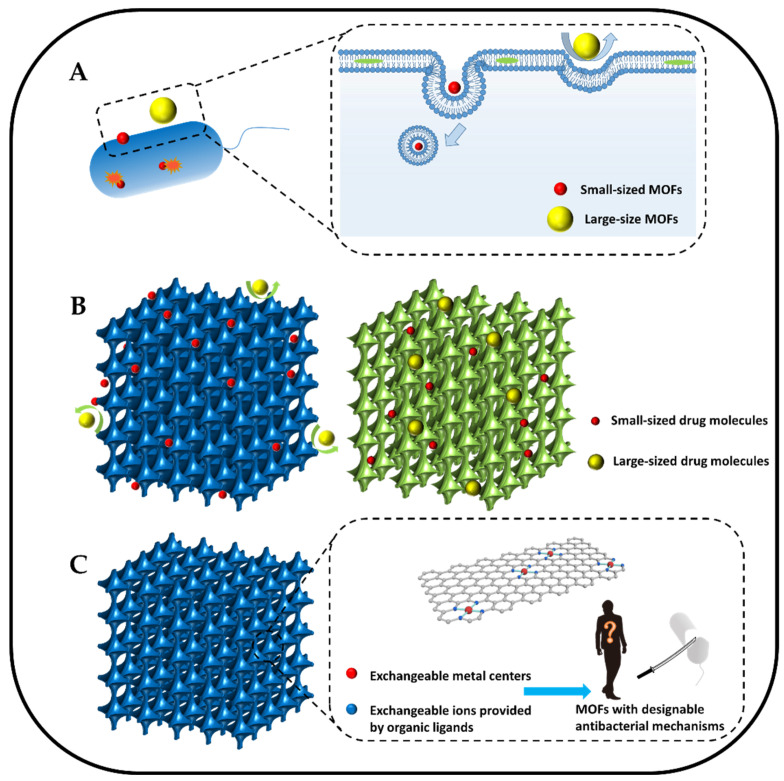
Schematic illustrations of strategies to enhance the antibacterial ability of MOFs: (**A**) the reduction in the size of MOFs can help the uptake of bacterial cells. (**B**) The increased pore size of MOFs can help load more large-sized drug molecules. (**C**) Alteration of the active sites of parent MOFs can confer new designable antibacterial mechanisms to daughter MOFs.

**Figure 5 jfb-13-00215-f005:**
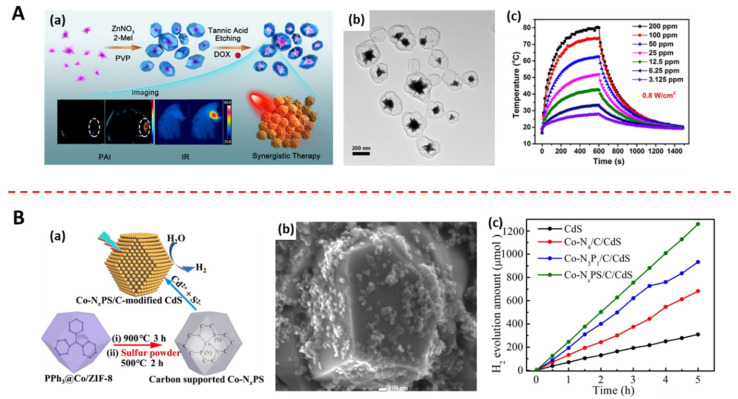
Methods of changing the structure of MOFs: (**A**) (**a**) Process of preparing a yolk−shell structure by selective etching of the ZIF−8 shell of Au@MOF by tannic acid (adapted from [146]); (**b**) TEM image of yolk−shell structured Au@MOF (adapted from [146]); (**c**) the temperature increasing and cooling curves of Au@MOF aqueous solution with various concentration under the irradiation of an 808 nm laser at 0.8 W cm^−2^ for 10 min (adapted from [146]). (**B**) (**a**) Process of preparing Co−N_X_PS/C/CdS, a heterojunction of CdS and a MOF with an altered active site (adapted from [147]); (**b**) SEM image of Co−NxPS/C/CdS (adapted from [147]); (**c**) H_2_ evolution amount graph of Co−NxPS/C/CdS and its precursors (adapted from [147]).

**Figure 6 jfb-13-00215-f006:**
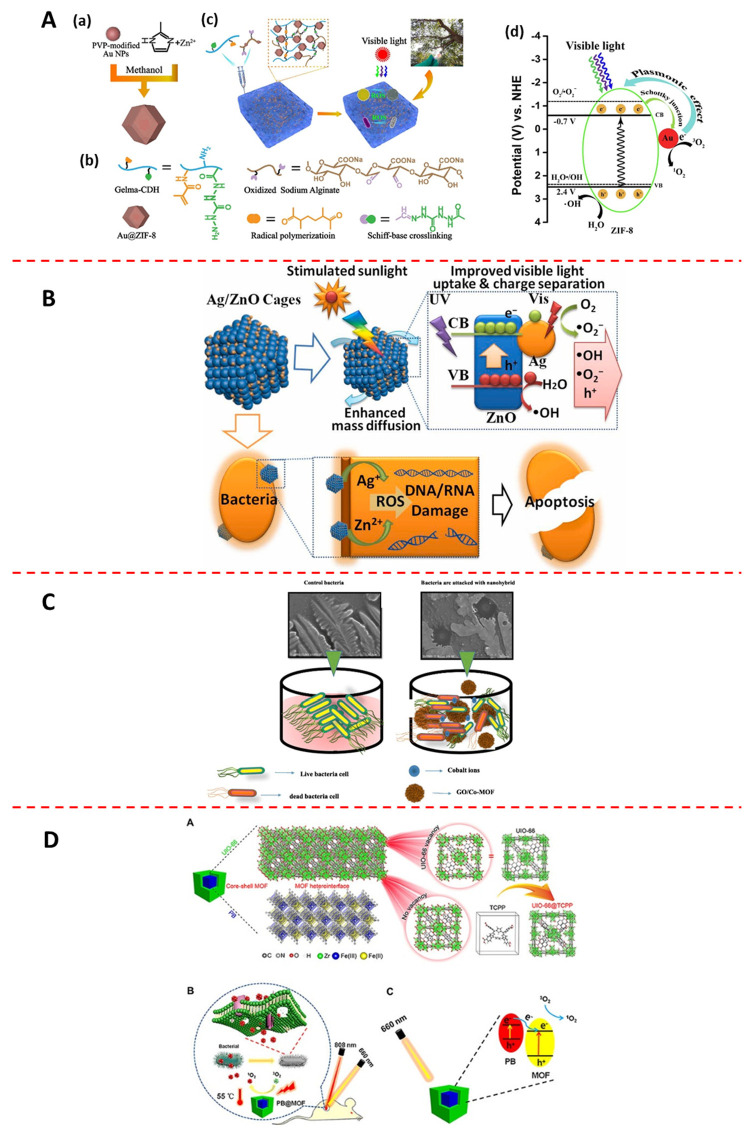
Enhancement of antibacterial ability of MOFs by constructing MOF-based composites: (**A**) (**a**) Process of preparing of Au@ZIF8, which possesses stronger PDT activity than ZIF−8 (adapted from [57]); (**b**) schematic illustration of infection model of Au@ZIF−8 injected into hydrogels for PDT therapy (adapted from [57]); (**c**) representative chemical structure of each component in hydrogels (adapted from [57]); (**d**) mechanism for ^1^O_2_ and ·OH generation under visible−light irradiation (adapted from [57]). (**B**) Schematic illustration of a heterojunction of Ag/ZnO with efficient PDT activity synergistically releasing the respective components for antibacterial purpose (adapted from [60]). (**C**) Go/Co−MOF synergizing the physical interaction of GO and the release of Co ions to achieve efficient antibacterial effect (adapted from [61]). (**D**) Schematic illustrations of PB@UIO−66−TCPP with synergistic PTT activity of PB and PDT activity of TCPP (adapted from [62]).

**Table 1 jfb-13-00215-t001:** Antibacterial activity of MOFs and MOF-derived materials.

MOF	Microbial Strain	Antibacterial Activity				Ref.
		Average Inhibition Diameter (mm)	MBC (μg mL^−1^)	MIC (μg mL^−1^)	Other Method	
ZIF-67	*S. cerevisiae* *P. putida* *E. coli*	15 15 15	–	– – 5		[49]
Co-SIM-1	*S. cerevisiae* *P. putida* *E. coli*	15 15 15	–	– – 5		
[(AgL)NO_3_]·2H_2_O	*E. coli* *S. aureus*	13 16	–	300 297		[50]
[(AgL)CF_3_SO_3_]·2H_2_O	*E. coli* *S. aureus*	15 16	–	300 307		
[(AgL)ClO_4_]·2H_2_O	*E. coli* *S. aureus*	15 19	–	308 293		
BioMOF-5	*S. aureus*	–	1700	4300		[51]
{[Zn(μ-4-HZBA)_2_]_2_·4(H_2_O)}_n_	*S. aureus*	–	–	–	a half maximal effective antibacterial concentration of about 20 mg L^–1^	[52]
ZIF-L	*E. coli* *S. aureus*	–	–	–	log reduction > 7 for *E. coli* and *S. aureus; SEM images*	[53]
CPPs	*B. subtilis* *P. vulgaris* *S. aureus* *P. aeruginosa* *S. enteritidis*	–	–	<25		[54]
Zn-PDA	*S. aureus* *B. subtilis* *A. baumannii* *K. pneumoniae* *S. entica* *E. coli*	17 16 11 9.7 9.7 8.6	–	300–308		[55]
Cu^2+^-doped PCN-224	*S. aureus*	–	–	–	antibacterial efficacy (99.71%)	[56]
Au@ZIF-8	*S. aureus* *E. coli*	–	–	–	inhibition ratio against *S. aureus* and *E. coli* was >99.9% when the dosage of Au@ZIF-8 was 0.2 mg mL^−1^	[57]
SPZA	*S. aureus* *E. coli*	–	–	–	The number of viable bacterial cells on SPZA is zero; SEM images	[58]
MnO_2_/ZIF-8	*E. coli*	–	3.24	–	complete inactivation against *E. coli* at low concentrations (3.24 μg mL^−1^)	[59]
Ag/ZnO	*E. coli* *S. aureus*	–	12.5 6.25	6.25 3.12		[60]
GO/Co-PTA	*E. coli* *S. aureus*	–	–	–	inhibited the growth of *E. coli* and *S. aureus* by up to 99%	[61]
Dual MOFs	*E. coli* *S. aureus*	–	–	–	the antibacterial efficiency against *S. aureus* and *E. coli* exceeded 99.31 and 98.68%, respectively.	[62]
ZGO-NH	*E. coli* *S. aureus*	2.59 3.82	-	-		[63]
ZIF-8@RFP	*MRSA*	-	10	-		[64]

**Table 2 jfb-13-00215-t002:** Antibacterial components and main roles of MOFs.

IBUs	OBUs	Functional Materials	Main Role of MOFs in Antibacterial Application	Ref.
Ag^+^	3−(Biphenyl−4−yl)−5−(4−tertbutylphenyl)−4−phenyl-4H−1,2,4−triazole	AgTAZ	Ag^+^ release	[49]
Ag^+^	Tris−(4−pyridylduryl)borane(L)	[(AgL)NO_3_]·2H_2_O	Ag^+^ release	[50]
Ag^+^	1−Butylimidazole	[Ag(Bim)]	Ag^+^ release	[78]
Co^2+^	Hmim	ZIF−67	Co^2+^ release	[49]
Cu^2+^	Trimesic acid	HKUST−1	Co^2+^ release	[74]
Cu^2+^	1,4−Benzendicarboxylic acid	Cu−SURMOF−2	Cu^2+^ release	[79]
Zn^2+^	Azathioprine	BioMOF−5	Zn^2+^ release	[51]
Zn^2+^	Hmim	ZIF−8	Organic compound generation	[80]
Ni^2+^	Hmim	Ni−Hmim	Hmin release	[77]
Zn^2+^	4−HZBN	[Zn(μ−4−HZBA)_2_]_2_·4(H_2_O)}_n_	HZBN release	[52]
Zr^4+^	BA, TCPP	VAN	VAN release	[81]
Fe^3+^	H_2_BDC	VAN	VAN release	[82]
Cu^1+^	1,3,5−Tribromobenzene + diethylamine → H_3_BTTri	GSNO	NO release	[83]
Zr^4+^	BA, TCPP	L−Arg	NO release	[84]
Hf^4+^	2−Hydroxyterephthalic acid, acetic acid	MnCO	CO release	[85]
Zn^2+^	Hmim	Pd(H)	H_2_ release	[86]
Fe^3+^	Acrylic acid	MIL−88B	CDT	[87]
Cu^2+^	1,3,5−Tricarboxybenzene	HKUST−1	CDT	[88]
Zr^4+^	H_2_BDC → I2−BDP	UIO−PDT	PDT	[6]
Hf^4+^	H_2_DBP	DBP−UIO	PDT	[89]
Hf^4+^	TCPP	Hf−TCPP NMOF	PDT	[90]
Zr^4+^	H_2_TCPP	PCN−224	PDT	[91]
Fe^2+^/ Fe^3+^	K_4_[Fe(CN)_6_]	PB	PTT	[92]
Mn^2+^	IR825	Mn−IR825	PTT	[93]
Zr^4+^	(Fc(COOH)_2_)	Zr−Fc MOF	PTT	[94]
Fe^3+^	GA	Fe−CPND	PTT	[95]
Zn^2+^	Hmim	ZIF−8 → PMCS	SDT	[96]
Zn^2+^, Mn^2+^	Hmim	ZIF−8 → DHMS	SDT	[97]
Zr^4+^ Pt^4+^ Au^3+^	BA	HNTM → HNTM−Pt@Au	SDT	[98]

**Table 3 jfb-13-00215-t003:** Composition and antibacterial mechanism of antibacterial agents for MOF-based composites.

Material 1	Material 2	Compound Mode	Main Role of MOFs in Antibacterial Application	Ref.
g-C_3_N_4_	Cu_3_P	Heterojunction	H_2_ loaded PTT PDT CDT	[136]
ZIF-8	Au NStar	Core–shell	Ag^+^ release PTT	[146]
PCN-224	Cu^2+^	stem grafting	PDT PTT	[56]
ZIF-8	Au NPs	Heterojunction	PDT	[57]
ZIF-8	Ag NPs	Heterojunction	Ag^+^ release PDT	[58]
ZIF-8	MnO_2_	Heterojunction	Ag^+^ release PDT	[59]
ZIF-8 → ZnO	Ag NPs	Heterojunction	Zn^2+^ release Ag^+^ release PDT	[60]
Ag-BTC-S/N	CQDS	Heterojunction	Ag^+^ release physical interaction CDT	[157]
Co-PTA	GO	Heterojunction	Co^2+^ release physical interaction	[61]
MIL-101	BQ	Core–shell	PDT PTT	[158]
UIO-66	PB	Core–shell	PDT PTT	[62]
multivariate MOF	Photosensitized porphyrin, BA	Core–shell	PDT	[159]
ZIF-8	-NH_2_	stem grafting	Zn^2+^ release	[63]
ZIF-8@RFP	o-NBA	Core–shell	Zn^2+^ release Antibiotic loaded	[64]
VAN@ZIF-8	FA	Core–shell	Zn^2+^ release Antibiotic loaded	[160]

## Data Availability

Not applicable.

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
