# Peer review of "MOFs and MOF-Derived Materials for Antibacterial Application"

_jfb, 2022, doi:10.3390/jfb13040215_

Round 1
Reviewer 1 Report
This review tries to summarize the outcome of a massive literature search on the antimicrobial effects of several types of MOFs and MOFs-based materials. The main benefit of this work are the two tables, which outline the correlation of the functional materials with the respective properties. The Authors made a huge effort to collect informations on several MOFs and their composition. Unfortunately, the outcame is a text which is very confused, with an excessive use of acronyms. There is not a clear flow and the overall result is that the focus on the antimicrobial activity of MOFs and MOFs-based materials is lost.
The figures are not explicative: in particular, the composite figures, namely fig. 3, 5 and 6, are a collection of images taken from the cited references, and are of poor quality (e.g., Fig. 3, panels B, C, E, F were clearly squeezed to fit the page size).
Last but not least, the English language needs significant improvements.
I am very sorry, but in my opinion, this review should not be published before an extensive reorganization and a careful language check.
Reviewer 2 Report
The current manuscript by Zhang, Peng and Wang represents an introductory review article on pathways for synthesizing MOFs for antimicrobial action. As a review article, it fails to meet part of its goals, and therefore I recommend it to be returned for a major review:
Major issues:
· The article fails to mention previous reviews on the topic - below are just three reviews with more than 500 citations in the literature that are not mentioned in the current manuscript. These articles should serve as a stepping stone for further development in the field rather than be represented slightly differently. Please emphasize this study's importance, as it is unclear why your review is being done and how it contrasts with reviews from the past two years.
[1] C. Pettinari, R. Pettinari, C. Di Nicola, A. Tombesi, S. Scuri, F. Marchetti, Antimicrobial MOFs, Coord. Chem. Rev. 446 (2021) 214121. https://doi.org/10.1016/j.ccr.2021.214121.
[2] M. Shen, F. Forghani, X. Kong, D. Liu, X. Ye, S. Chen, T. Ding, Antibacterial applications of metal–organic frameworks and their composites, Compr. Rev. Food Sci. Food Saf. 19 (2020) 1397–1419. https://doi.org/10.1111/1541-4337.12515.
[3] G. Wyszogrodzka, B. Marszałek, B. Gil, P. Dorozyński, Metal-organic frameworks: Mechanisms of antibacterial action and potential applications, Drug Discov. Today. 21 (2016) 1009–1018. https://doi.org/10.1016/j.drudis.2016.04.009.
§ The manuscript brings a collection of more than 160 citations that are not summarized in a way that allows a direct comparison. A review article should not be just a mechanistic collection of articles. A review article should advance the understanding in the field. This could be done by summarizing the efficiency of the best MOFs for given types of bacteria (fungi, etc.) or comparing the efficiency of a MOF prepared via different methods.
Minor issues:
§ Figures and tables should be put close to their first mention in the text.
§ Citations should include the year or reference number at first mention. For example, Li et al 2009 or Li et al. [400], instead of adding the reference number in the end of the paragraph.
§ The language style is heavy and requires sentence shortening. As an illustrative example, lines 78-86 contain two sentences that are four rows each, the same for lines 42-46, 92-96, and so on.
§ Figures suffer from aspect ratio distortion (e.g. Fig. 3E-F).
Reviewer 3 Report
Dear Authors,
Thank you for your interesting and comprehensive review! I believe it can be useful for those researchers who need to touch the area of MOFs. Nevertheless, I have some comments on it:
1. Since there are other prospective biomimetic technological routes to construct antibacterial drugs, including the protein cages (e.g. ferritin-based) and cell-membrane-camouflaged vesicles, I recommend to add a comparative section to stress the MOFs pros and contras in comparison with the other technologies.
2. The Figures 3-6 captions have to contain a generalizing part, e.g. "Figure 3. Schematic illustrations for various ZIF-constructions: (a) ZIF-L with nano-dagger morphology...".
3. The manuscript contains some typos, e.g.
Line 190, "3.1 physical interaction";
Line 202, "CH3COOH";
Line 260, "HZBA)2]2·4(H2O)}n)".
4. The text in the Figures 2, 3, 6 is unreadable and has to be enlarged.
5. The abbreviations in the Tables 1 and 2 should be given additionally in the footnotes.
6. The References have to be reformatted according the the MDPI style.
Round 2
Reviewer 1 Report
The text is really improved. It is a very useful resource for MOF and MOF related research field. Thank you to the authors for this interesting review.
Author Response
Thank you very much for your kind words about our revised manuscript.
Reviewer 2 Report
In the revised version of the manuscript, the authors address all my questions and concerns in detail. They have added additional literature and discussed their review in light of previous studies. They have included other summary tables and an abbreviation list. The authors also extensively edited the text and the Figures to make them more readable and easier to perceive. I recommend the manuscript for publication as it is.
Author Response
We sincerely thank you for your kind words about our revised manuscript.
Reviewer 3 Report
Dear Authors,
Thank you for addressing all my comments! I think the manuscript can be accepted in the present form.
Author Response

(The authors gave the same response as above.)
